# Biceps Tenodesis Versus Tenotomy with Fast Rehabilitation Protocol—A Functional Perspective in Chronic Tendinopathy

**DOI:** 10.3390/jcm9123938

**Published:** 2020-12-04

**Authors:** Jan Zabrzyński, Gazi Huri, Szymon Gryckiewicz, Rıza Mert Çetik, Dawid Szwedowski, Łukasz Łapaj, Maciej Gagat, Łukasz Paczesny

**Affiliations:** 1Department of General Orthopaedics, Musculoskeletal Oncology and Trauma Surgery, University of Medical Sciences, 61-712 Poznań, Poland; esperal@o2.pl; 2Citomed Healthcare Center, Department of Orthopaedics, Orvit Clinic, Sklodowskiej 73, 87-100 Toruń, Poland; szymon.gryckiewicz@wp.pl (S.G.); dszwedow@yahoo.com (D.S.); drpaczesny@gmail.com (Ł.P.); 3Department of Histology and Embryology, Faculty of Medicine, Collegium Medicum in Bydgoszcz, Nicolaus Copernicus University in Torun, 85-067 Bydgoszcz, Poland; mgagat@cm.umk.pl; 4Orthopaedics and Traumatology Department, Hacettepe Universitesi, 06100 Ankara, Turkey; gazihuri@yahoo.com (G.H.); rmcetik@hacettepe.edu.tr (R.M.Ç.); 5Orthopaedic Arthroscopic Surgery International (OASI) Bioresearch Foundation, 20133 Milan, Italy

**Keywords:** LHBT, tenodesis, tenotomy, tendinopathy, biceps, rehabilitation, arthroscopy, shoulder

## Abstract

The study aimed to evaluate the results after arthroscopic tenodesis and tenotomy of the biceps tendon (LHBT), coupled in tenotomy modality with a personalized postoperative rehabilitation protocol. The study included patients who underwent arthroscopic biceps tenotomy or tenodesis due to chronic biceps tendinopathy. Postoperatively, a standard rehabilitation program was prescribed to the tenodesis group and personalized was introduced in the tenotomy group, respectively. The outcomes were assessed using the American Shoulder and Elbow Surgeons scale (ASES), clinical tests that are dedicated to biceps tendinopathy, the occurrence of a Popeye deformity, night pain, and return to previous sporting activities. A cohort of 67 patients was enrolled in the final follow-up examination (mean 27 months) of which 40 patients underwent tenotomy (60%), and 27 patients (40%) underwent tenodesis. The mean ASES score improved from 48.1 to 87.8 in the tenotomy group and from 44 to 72.7 in the tenodesis group during the follow-up (*p* < 0.0001). The tenotomy group had better mean postoperative ASES scores than the tenodesis group (*p* < 0.0001). Positive clinical tests for biceps pathology were noticed more often in patients after LHB tenodesis (*p* = 0.0541). The Popeye deformity occurred more often in the tenotomy group; however, no patient complained of the visual appearance of the arm contour (*p* = 0.0128). Moreover, the frequency of night pain decreased in the tenotomy group (*p* = 0.0059). Return to previous sporting activities was more frequent in the tenotomy group (*p* = 0.0090). Arthroscopic biceps tenotomy is a reproducible, simple procedure, and augmented by a rapid rehabilitation protocol that provides promising clinical outcomes, reduces shoulder pain, and allows the patient to return to previous sporting activities, even in population older than 50 years.

## 1. Introduction

Tendinopathy is a chronic disorder of a tendon with dysregulation of its immune response, which leads to failed tendon healing with an abundance of degenerative processes and marginal inflammatory infiltration [1,2]. Thus, treatment of tendinopathy is often challenging, and there is no ideal method of treatment. The long head of the biceps (LHBT) tendinopathy is usually associated with repetitive, chronic traction and friction forces affecting the tendinous tissue as a microtrauma during shoulder movement [3,4,5]. The LHBT anatomy is unique since the tendon consists of two different portions: the intra-articular portion and the extra-articular portion [6]. Passing underneath the rotator cuff (RC) to its anchor on the superior labrum and glenoid, the intra-articular part is more prone to trauma, especially that associated with the rupture of neighboring structures, such as rotator cuff tears (RCTs), superior labrum anterior to posterior tears (SLAP), or subacromial bursa inflammation leading to subacromial impingement (SI) [3,7,8,9].

A significant part of the population suffers from anterior shoulder pain, and the LHBT pathology is an undeniable source of this condition [6,10]. The LHBT often compensates for the abnormal forces in the overloaded, painful shoulder. Thus, its tendinopathy is usually a secondary phenomenon, related to severe concomitant shoulder disorders [7].

The pathology affecting the LHBT has been well known and recognized since the 19th century; however, the treatment of chronic, persistent biceps tendinopathy remains controversial [8]. The development of shoulder arthroscopy allowed for the exploration and detailed investigation of the joint and its pathologies, and for simultaneous development of modern treatment methods. The two main surgical procedures are tenotomy and tenodesis; performed either open or arthroscopy-assisted [11,12,13,14]. Biceps tenotomy and tenodesis result in high patient satisfaction and pain relief and moreover, as Aflatooni et al., showed, patients in both groups reported with equal frequency (95%) that they will repeat the surgery [12,14].

Recent reports about modified techniques of LHBT tenotomy with superior functional outcomes are promising [11]. However, due to confirmed, similar results after these two surgical procedures, we decided to improve the routine rehabilitation protocol, recommended after arthroscopic tenotomy procedure, given that such exercises are not put in the unified schedule, to obtain better outcomes and ensure faster return to previous sports activity.

The study aimed to evaluate the results after arthroscopic tenodesis and tenotomy of the biceps tendon, augmented in tenotomy with a personalized postoperative rehabilitation protocol.

## 2. Experimental Section

The retrospective sequential series study was approved by the Bioethics Committee of the Nicolaus Copernicus University in Toruń functioning at Collegium Medicum in Bydgoszcz (approval no. KB 62/2017). All patients were volunteers who received oral and written information about the study and gave informed consent. The study was conducted in compliance with The Code of Ethics of the World Medical Association (Declaration of Helsinki) for experiments involving humans.

The study included 81 patients who underwent treatment in the Orthopedic Department specializing in arthroscopic and mini-invasive surgery. The inclusion criteria were: patient age ≥25 and <79 years, proximal biceps tendinopathy (tears, instability, SLAP tears) diagnosed preoperatively with physical examination and imaging techniques (magnetic resonance, sonography) and confirmed during diagnostic arthroscopy as well as a history of no improvement after at least six months of conservative treatment. A minimum of six months follow-up was also required for inclusion.

Of the 81 patients included in the study, three of them were lost during the follow-up as they moved from their previous addresses, whereas 11 were unavailable for the final follow-up and did not respond to contact. The data comprising the age, gender, sports activity and the American Shoulder and Elbow Surgeons Score (ASES) were recorded preoperatively.

All patients underwent shoulder arthroscopy (by using a standard 30° arthroscope, Smith&Nephew, Memphis, TN, USA) in the beach-chair position and had a diagnostic arthroscopy performed with LHBT tenodesis or tenotomy with the repair of any concomitant lesions. The patients based on surgeon experience and patients preoperative qualification were assigned to the tenodesis/tenotomy group. The patients were informed on the possible complications and step-by-step surgical procedure with subsequent rehabilitation protocol. The standard posterior portal and additional working portals were used. For the tenodesis procedure, the bicipital groove was prepared with a curette to encourage bone-tendon healing, and the suture anchor (4.5 mm double-loaded suture anchor, Twinfix, Smith&Nephew, Memphis, TN, USA) was fixed in the center of the bicipital groove. Sutures were passed through the LHBT and tied, afterwards the tendon was cut above the suture anchor. Subsequently, the LHBT was released near its insertion at the superior labrum, and the biceps stump was removed from the shoulder using a grasping tool. For tenotomy, the LHBT was first cut parallel to its insertion to the labral complex; subsequently, the superior labrum was debrided. This technique provides an extensive biceps stump, which results in its entrapment in the area of the biceps pulley in some patients.

The rehabilitation protocol after the tenotomy procedure included an arm sling used for three weeks, early passive range of motion (PROM) exercises and, after the first week, active shoulder range of motion (AROM) exercises (Table 1).

Active flexion of the elbow was allowed in the third week postoperatively. The rehabilitation protocol after the tenodesis procedure included an arm orthosis used for five weeks, and passive shoulder range of motion exercises was followed by active exercises from the fifth week onwards (Table 1). About 90% of patients participated in exercises with physiotherapists, while 10% were precisely instructed about the rehabilitation program. However, if there was a concomitant pathology, i.e., RCTs, the rehabilitation was adjusted accordingly. In patients with a supraspinatus (SST) tear or massive RCT (with SST retraction), we recommended the use of an arm abduction orthosis.

Patients met regularly with physiotherapists, especially during the first weeks of rehabilitation, connected with working on passive and assisted movements (Figure 1, Figure 2 and Figure 3). The main parameters and goals in rehabilitation were pain-free PROMs, AROMs, and good muscle strength. Additional aspects of rehabilitation were to improve the quality of movement (correct thoraco-scapular rhythm and scapula static position) and the right muscle balance (between different groups).

The follow-up examination included outcome assessment utilizing physical examination, ASES scoring, clinical tests dedicated for biceps tendinopathy (Speed test, tenderness over the bicipital groove test, Yergason test), the occurrence of the Popeye deformity (PD) (enlarged bulging of muscle observed on the distal biceps region), the occurrence of night pain, other complications, and return to previous sporting activities after the rehabilitation period.

The data were analyzed with the use of GraphPad Prism v.8.0.1 (GraphPad Software, La Jolla, CA, USA). *p* < 0.05 was considered significant. Variables were tested for normality by the Shapiro–Wilk test. Correlation of variables within patients was conducted with the Spearman Rho correlation coefficient. The Mann–Whitney U and Wilcoxon tests were applied to compare the data. The Fisher’s exact test was used to compare descriptive characteristics. The comparison between groups and statistical analysis were done blinded and by an independent investigator.

## 3. Results

The mean age of the study group was 55 years (range: 27–75; SD: 10.7), while gender distribution was 25 women to 42 men. The history of recreational sports activity at least once a week before the onset of symptoms was reported by 43 patients (64%) (Figure 4A).

In the examined cohort, 40 patients underwent tenotomy (60%) whereas 27 patients (40%) underwent tenodesis; moreover, a total of 64 out of 67 patients (96%) had additional procedures performed during the surgery: 41 patients underwent RCT repair (16 massive tears and 25 tears of the single tendon), one patient had chronic RC tendinopathy with calcium deposits, 7 patients had various types of SLAP repairs, 7 patients had ACJ resections, 12 had subacromial bursa resections, 3 patients had anterior labrum repairs, and 4 patients had joint debridement due to omarthrosis (OA) (Figure 4B). There were only three patients with isolated LHBT disorders. The patients undergoing arthroscopic tenodesis (mean age 51, range: 27–75, SD—11.5) were statistically younger than those undergoing tenotomy (mean age 58, range: 30–74, SD: 9.3), (*p* = 0.0029) (Figure 4C). The average period of follow-up in the cohort was 27 months (range: 7 to 54 months; SD: 13.2). There were no wound infections or reoperations reported during the follow-up period. The comparison of descriptive data from the follow-up examination was summarized in Figure 5.

The Popeye deformity occurred more often in the tenotomy group than in the tenodesis group. However, no patient complained about the visual appearance of the arm contour (*p* = 0.0128) (Figure 5A). Moreover, the frequency of night pain decreased in the tenotomy group compared to the tenodesis population (*p* = 0.0059) (Figure 5B). Persistent bicipital groove tenderness in clinical tests was noticed more often in patients after LHB tenodesis than in those after tenotomy (*p* = 0.0541) (Figure 5C). Return to previous sporting activities was more frequent in the tenotomy group (*p* = 0.0090) (Figure 5D).

The mean ASES score improved from 46 preoperatively (range: 34–60) to 80 postoperatively (range: 50–100) in the studied cohort (*p* < 0.0001) (Figure 4D). The mean ASES score improved from 48.1 (range: 34–60) to 87.8 (range: 50–100) in the tenotomy group and from 44 (range: 34–60) to 72.7 (range: 50–98) in the tenodesis group during the follow-up (*p* < 0.0001 and *p* < 0.0001, respectively) (Figure 4E,F). The tenotomy group had better mean postoperative ASES scores than the tenodesis group (*p* < 0.0001) (Figure 4G). There was no correlation between age and pre/postoperative ASES (*p* = 0.5020 and *p* = 0.3662, respectively) (Figure 4H,I). The mean follow-up period and postoperative ASES revealed a positive correlation in the studied cohort (rho = 0.307, *p* = 0.01155) (Figure 4J). Females and males had similar mean preoperative ASES: 46.8 (range: 34–60) and 46.1 (range: 34–60), respectively (*p* = 0.7241); however, males had a better final score: 83.9 (range: 50–100) than females: 78 (range: 58–100) with no statistical differences (*p* = 0.1519) (Figure 5E,F). The improvements in mean pre/postoperative ASES in both females and males were statistically significant (*p* < 0.0001).

Moreover, the patients with a history of sporting activities had better mean postoperative ASES compared to the patients without routine participation, 87.9 (range: 50–100) vs. 70.6 (range: 50–100), respectively (*p* < 0.0001) (Figure 5G). The mean postoperative ASES in patients with the Popeye deformity complication: 88.6 (range: 50–100), was not statistically significant when compared to the non-Popeye deformity group: 80.2 (range: 50–100) (*p* = 0.0934) (Figure 5H). Patients with persistent night pain had lower mean postoperative ASES than those without this complication occurrence: 69.6 (range: 50–98) vs. 85.2 (range: 50–100); (*p* = 0.0003) (Figure 5I). 

## 4. Discussion

Though the initial treatment of tendinopathy is non-operative, it is usually recommended for three to six months as the first-line treatment [3,7,8]. The rest on the arm sling, nonsteroidal anti-inflammatory drugs (NSAID) and corticosteroid injections, free-handed or ultrasound-guided, and physical therapy (kinesiotherapy: stretching, scapular muscle strengthening, restoration of the full range of motion in the glenohumeral joint, improvement in the movement pattern) are commonly applied [15]. Krupp et al., introduced a rehabilitation protocol for biceps tendinopathy which starts with pain management, the restoration of full passive range of motion (ROM), then active ROM and, finally, strength training [16]. It has been demonstrated that surgical intervention is an appropriate therapy for symptomatic LHBT disorders, especially after three months of initial, unsuccessful conservative treatment. The abundance of our surgically-treated cohort is proof that LHBT tendinopathy is usually an insidious and difficult pathology that does not always respond to conservative treatment.

The authors reported simultaneous clinical outcomes after biceps tenotomy and tenodesis. The authors of this study augmented the postoperative protocol with a rapid rehabilitation program to obtain better clinical results. Proceeding in compliance with this program allowed us to improve outcomes, especially in the tenotomy treated group. In 1990, Patte et al., noted the reduction of shoulder pain after spontaneous rupture of the LHBT, afterwards, Walch et al., proposed arthroscopic tenotomy as a palliative treatment method in patients with irreparable RCTs [6,17]. Additionally, authors recommend LHB routine tenotomy in elderly patients with significant biceps abnormality during shoulder arthroscopy. Kempf et al., noticed better clinical outcomes in the group that underwent routine tenotomy of the LHB with RCT repair [18]. Although tenotomy is a simple procedure, an important fact is that it is not suitable for every patient and that it demands an individual approach. On the other hand, this procedure provides a significant alleviation of pain during movement and reduces tenderness during palpation of the bicipital groove [19]. Gill et al., reported a group of 30 patients who underwent arthroscopic biceps tendon release for the treatment of LHBT pathology: 97% of patients did not require additional NSAIDs at follow-up, while 90% of the patients returned to their previous activity [20]. In our cohort, even though the patients who underwent tenotomy were older than those from the tenodesis group, the reduction of pain after tenotomy was reflected by a high ASES score, mostly negative clinical tests for the LHBT pathology, and rare persistent night pain. Gill et al., reported the mean postoperative ASES score at 81.8, Kelly et al., at 75.6, and the score in our cohort improved and reached a mean value of 87.8, probably due to the introduction of the postoperative rehabilitation protocol [19,20].

The cosmetic arm deformity, referred to as “a positive Popeye deformity” (abnormal shortening and cosmetic defect of the belly of the biceps muscle), is the most frequent complaint after biceps tenotomy. Kelly et al., revealed the Popeye deformity in 70% of the cases in a group of 54 patients after LHB tenotomy [19]. In our cohort, the PD was present in less than half of the patients after tenotomy, and the possible cause could be iatrogenic—excessive rehabilitation or failed tendon stump fixation. Complications challenge orthopedic surgeons to invent new methods of LHBT pathology treatment. Narvani et al., described a novel technique for LHB tenotomy that avoids the Popeye muscle deformity: the authors specially incise the tendon, thus creating a kind of anchor in the bicipital groove [11]. We connect the reduction of Popeye deformity with the use of “anchoring” like the LHB tenotomy technique [21]. Osbahr et al., cut the LHBT near its anchor to the labrum, which probably led to a trap in the bicipital groove, the so-called autotenodesis phenomenon [21,22]. Although we also prefer this method of performing tenotomy, we observed no differences in ASES score between the Popeye and non-Popeye groups, despite the presence of the cosmetic deformity.

On the other hand, biceps tenodesis consists of tenotomy and anchoring the proximal part of the tendon to a certain location: the bony area of the bicipital groove, soft tissue or a transfer to the coracoid process. Biceps tenodesis is usually recommended for younger (<50 years old) and physically active patients (athletes and heavy workers). Additionally, tenodesis allows for preserving the normal contour of the arm [6]. The LHBT tenodesis procedure may be associated with complications: persistent bicipital groove pain, failed fixation with subsequent Popeye deformity, humeral shaft fractures at the site of fixation, and a limited range of motions of the affected glenohumeral joint [23,24,25]. The authors commonly report good or excellent results of tenodesis in the groups they treated. In the groups studied by Duchman et al., and Friedman et al., the mean ASES scores were 81.6 and 85.2, respectively [14,23]. The mean ASES score in our tenodesis-treated group was 72.7 with more frequent positive painful clinical tests for LHB tendinopathy, night pain and reduced return to previous sporting activities, which may be connected with implant presence inside the bicipital groove. Moreover, we have noticed some cases of Popeye deformity after arthroscopic tenodesis. These results also may be linked to the postoperative rehabilitation program. The rehabilitation protocol after tenodesis is not as simple and fast as it is in tenotomy and may generate some problems. The general rehabilitation protocol assumes the performance of exercises aimed at the progressive improvement of the range of motion of the glenohumeral joint during the first six weeks with an additional restriction of elbow flexion/extension and supination/pronation [6]. This conservative rehabilitation protocol may lead to limited ROM after the surgery and poor ASES score; on the contrary, tenotomy allows for early rehabilitation, thus the authors of this study agreed to a fast rehabilitation program starting with assisted active movement in the first week postoperatively. Limitations in the glenohumeral joint movement can restrain the return to previous sports activity. On the other hand, starting AROMs earlier under the tenotomy protocol may lead to the destabilization of the trapped LHBT inside the bicipital groove and increase the Popeye deformity rate. A customized program with a physiotherapist allows a reduction in the risk of complications.

To increase ROM and muscular strength after arthroscopy, it is important to consider the position and kinematics of the scapula. Scapular position and kinematics are important factors that can affect patient symptoms [15,26]. Scapular muscles (serratus anterior, trapezius) are as important for the shoulder as rotator cuff muscles [27,28]. Motivation and cooperation with the patient during the rehabilitation process can influence the final results. The longer follow-up in our cohort correlated positively with the postoperative ASES score and it may be linked to the longer period of cooperation with a physiotherapist. Moreover, the sport-active group had better functional outcomes, which is probably based on improved preoperative musculoskeletal system efficiency and motivation.

A significant group of patients suffer from anterior shoulder pain due to the LHBT tendinopathy; however, this kind of tendinopathy is usually associated with concurrent shoulder pathologies, which significantly complicates the diagnostic process and treatment [6,7,8]. Similarly, most patients in the cohort we examined had a concurrent pathology of the affected shoulder with a strong predominance of RCTs. This is the main limitation of our study, predominantly since the fact that we included a group of patients with a combination of concomitant shoulder lesions, but, as we pointed out, it is characteristic and typical of LHB tendinopathy [6]. There were only three patients in the studied population with isolated biceps tendinopathy. We are aware that the tenotomy group had 24/40 (60%) RCTs with massive injuries and the tenodesis group had 16/27 (59%) of the same type injuries, but the percentage is almost the same. Moreover, probably the population with concomitant RCTs could have the largest impact on the postoperative outcomes. Complex injuries have an influence on and complicate the recovery and rehabilitation protocol; thus, we tried to adjust the protocol to RCT repair, SLAP repair, ACJ resection, anterior labrum repair and omarthrosis separately. Furthermore, the patients were assigned non-randomly, into the two treatment groups, and it was a retrospective case-series study design, thus we realize from the risk of bias.

## 5. Conclusions

Arthroscopic biceps tenotomy is a reproducible, simple procedure, and, augmented by a rapid rehabilitation protocol, provides promising clinical outcomes, reduces shoulder pain, and allows the patient to return to previous sporting activities, even in a population older than 50 years. A higher rate of Popeye deformity occurrence was not linked to impaired functional outcomes and was well tolerated by the patients. The fast rehabilitation protocol introduced after biceps tenotomy allowed us to gain better outcomes than after the tenodesis procedure with standard rehabilitation protocol.

## Figures and Tables

**Figure 1 jcm-09-03938-f001:**
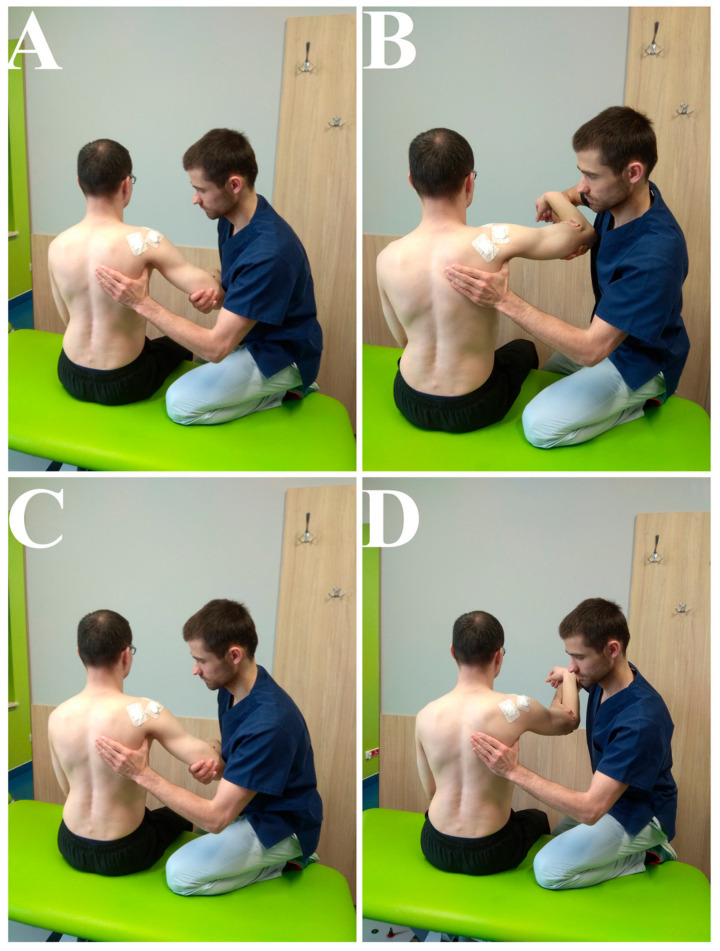
(**A**,**B**)—PROM exercises, abduction in GH (tolerated range: 0–90). (**C**,**D**)—PROM exercises, flexion in GH (tolerated range: 0–90).

**Figure 2 jcm-09-03938-f002:**
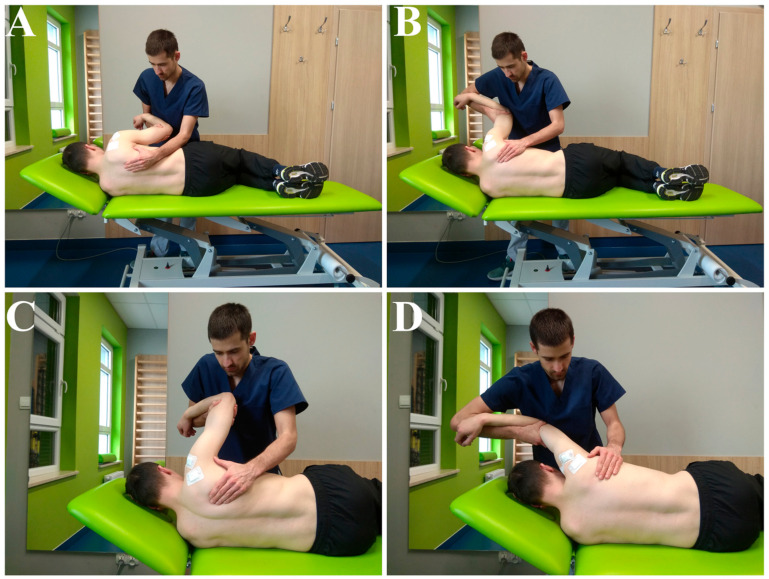
(**A**,**B**,**D**): PROM/AAROM, flexion in GH and SC (tolerated range: 0–180) (**C**): PROM/AAROM, abduction in GH and SC (tolerated range: 0–90).

**Figure 3 jcm-09-03938-f003:**
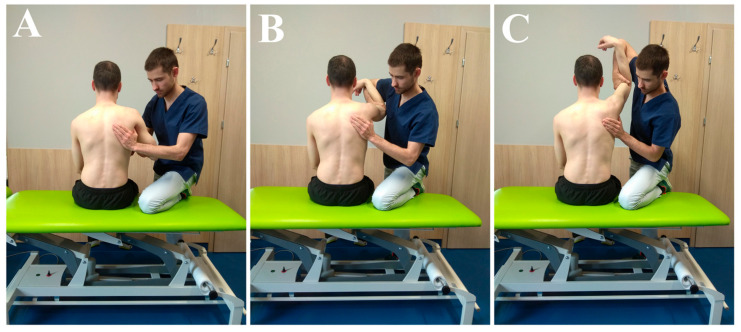
(**A**–**C**): AAROM/AROM, flexion in GH and SC (tolerated range: 0–180).

**Figure 4 jcm-09-03938-f004:**
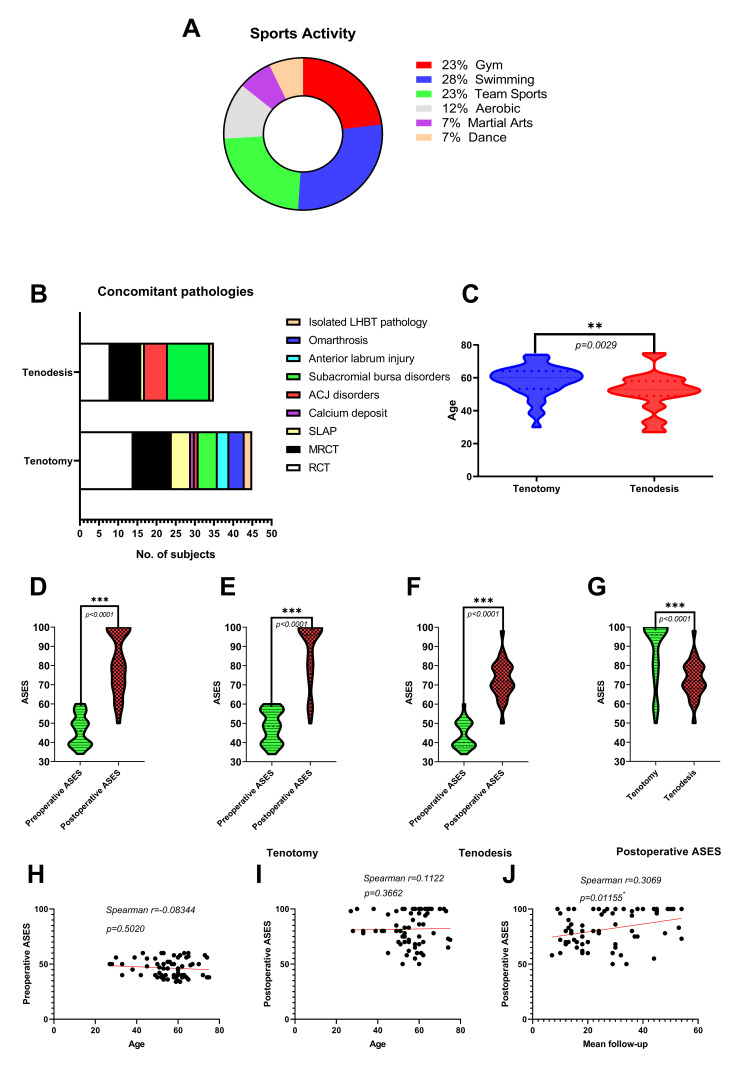
(**A**) The history of recreational sports activity of population (n = 43). (**B**) Concomitant disorders to LHB tendinopathy. (**C**) Comparison of patients’ age in tenotomy/tenodesis groups. (**D**) Comparison of mean pre/postoperative ASES score in population. (**E**,**F**) Comparison of mean pre/postoperative ASES score in tenotomy and tenodesis groups. (**G**) Comparison of mean postoperative ASES score between tenotomy and tenodesis groups. (**H**,**I**) Correlation between age and pre/postoperative ASES. (**J**) Correlation between mean follow-up and postoperative ASES.

**Figure 5 jcm-09-03938-f005:**
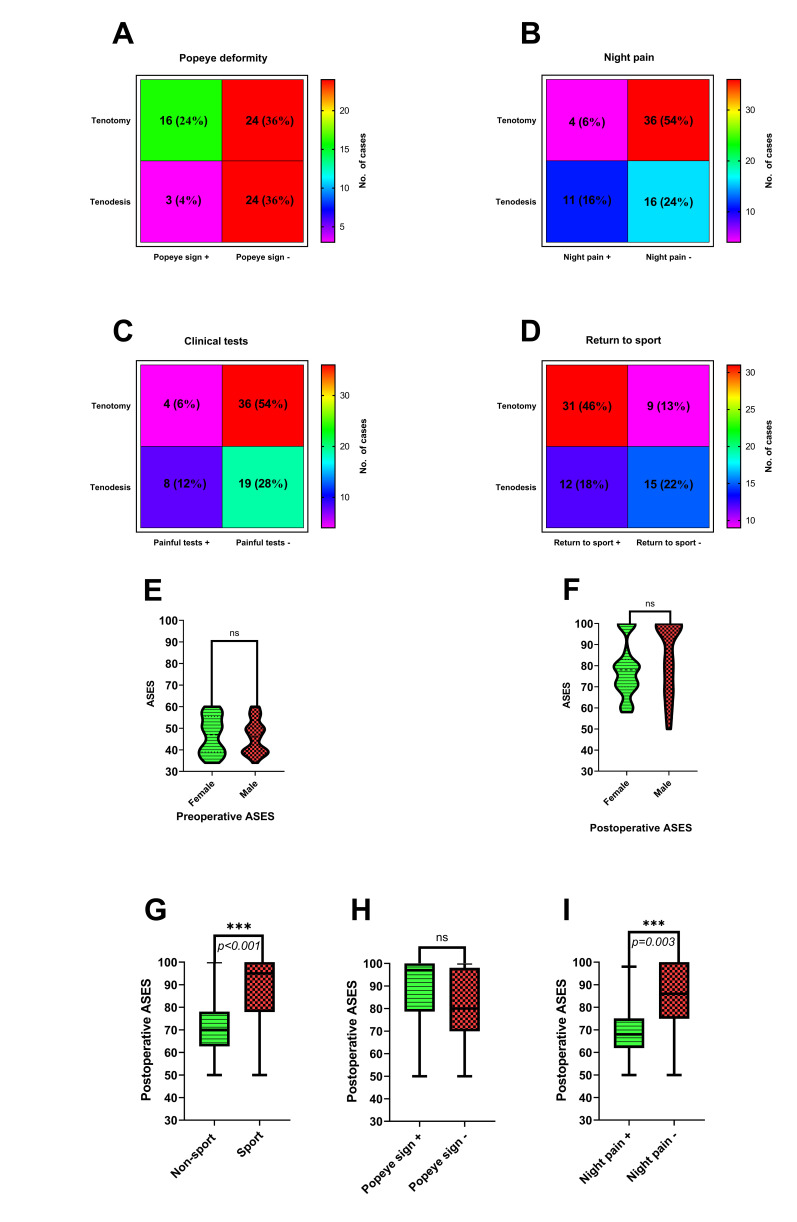
(**A**–**D**) Comparison of the descriptive data in tenotomy and tenodesis groups. (**E**,**F**) Comparison of mean pre/postoperative ASES score in females and males. (**G**–**I**) Comparison of mean postoperative ASES score depending on history of routine participation in sporting activities, Popeye sign occurrence and presence of night pain.

**Table 1 jcm-09-03938-t001:** The rehabilitation protocols in tenotomy and tenodesis groups.

Surgical Procedure	Tenotomy	Tenodesis
Brace/sling	3 weeks	5 weeks
1 week	PROM–GH&SC	PROM–GH&SC
	Isometric exercise	Isometric exercise
	AAROM	
2–3 week	AROM GH&SC	AAROM GH&SC
	Active elbow flexion/extension (from 3th week)	Passive elbow flexion/extension
4–5 week	Physiotherapy continuation	AROM GH and SC
		Active elbow flexion/extension, forearm pronation/supination (from 5th week)

(PROM—passive range of motion exercises; AROM—active shoulder range of motion exercises; AAROM—assisted active shoulder range of motion exercises; GH—Gleno-humeral joint; SC—Scapula).

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
