# Peer review of "Biceps Tenodesis Versus Tenotomy with Fast Rehabilitation Protocol—A Functional Perspective in Chronic Tendinopathy"

_jcm, 2020, doi:10.3390/jcm9123938_

Round 1

Reviewer 1 Report

The Methods in this paper are highly questionable:

  • The rate of lost to follow up is > 10 % of the study population.
  • The assignment of the patients into one or the other treatment cohort has not been described.
  • The influence of concomitant injuries concerning the correlation to the outcome is not displayed.
  • The postoperative protocol of the two treatment groups is different, e.g. not comparable.
  • Conclusions drawn from this data, while comparing different treatment modalities within the same group is not reasonable.

Therefor the manuscript should not be considered for publication.

Author Response

Dear Editor,

Thank you for the opportunity to improve and resubmit our manuscript entitled:

“Biceps tenodesis versus tenotomy with fast rehabilitation protocol – a functional perspective in chronic tendinopathy”

The suggestions offered by the reviewers have been immensely helpful. We appreciate all the comments on the manuscript.

We have included the reviewer comments, and responded to them individually, indicating how we addressed each concern and describing the changes we have made. The revised manuscript has been read and approved by all the authors. 

We wish to express again our appreciation for the insightful comments which have helped us significantly to improve our manuscript. 

Yours sincerely,

Jan Zabrzyński

Overall corrections:

  • English language was improved

Reviewer 1:

  • The rate of lost to follow up is > 10 % of the study population.

Thank you for the comment. Of the 81 patients included into the study, three were lost during the follow-up as they moved from their previous addresses, whereas 11 were unavailable for the final follow-up and did not respond to contact, and we accented these facts in the Methods section. We were unable to predict that > 10 % of the study population will not be able to participate in the follow up, which actually last for mean 27 months.

  • The assignment of the patients into one or the other treatment cohort has not been described.

Thank you for the valuable comment. We fully agree with the reviewer and indeed this fact was not particularly described. There were two arthroscopic surgeons, JZ and ŁP, and the first one performed 27 LHBT tenodesis and the second one performed 40 tenotomies of the final follow up population. These two surgeons are trained in mini-invasive arthroscopic procedures, and the methods they chose, are based on their experience. Moreover, during the qualification of patients for an arthroscopic treatment, surgeons explained to the patient, step-by-step, the surgical procedure and the possible complications, however, they always told the patients about the preferable arthroscopic technique. Reassuming, the patients were cohort non-randomly into the tenodesis/tenotomy group, and this is the reason why tenodesis patients were statistically younger than those undergoing tenotomy.

  • The influence of concomitant injuries concerning the correlation to the outcome is not displayed.

Thank you for the valuable comment. I would like to point that this issue was well explained in the limitations section. The LHBT tendinopathy is inevitably linked with the concurrent shoulder pathologies, due to anatomy, biomechanics and pathology. Of course, this fact significantly complicates the diagnostic process and treatment. The concomitant pathologies were displayed in the Fig. 4B, and as you can see, the RCTs and MRCTs occupy almost half of the investigated tenotomy/tenodesis groups.

  • The postoperative protocol of the two treatment groups is different, e.g. not comparable.

Thank you for the valuable comment. Indeed, the rehabilitation protocols were different, and this was an aim of our study. We evaluated the results after arthroscopic tenodesis and tenotomy of the biceps tendon, which mostly in literature are comparable, but still the superiority of one of these methods is debated. That is why, we performed the intensive rehabilitation, postoperatively, with a personalized protocol, to obtain better results in the tenotomy group. We wanted to show that this is possible from the functional perspective, and the complication such as Popeye deformity is not a clinically important problem. Moreover, the return to sport is also possible after the tenotomy procedure.

  • Conclusions drawn from this data, while comparing different treatment modalities within the same group is not reasonable.

Thank you for the valuable comment. We answered to this question in the subsections above.

Reviewer 2 Report

JCM 974334 - Biceps tenodesis versus tenotomy with fast rehabilitation protocol – a functional perspective in chronic tendinopathy

The authors aimed to evaluate the clinical results of their cohort of 67 patients who underwent either an arthroscopic tenodesis (27 patients) or a tenotomy (40 patients) of the long head of biceps tendon (LHBT) for chronic biceps tendinopathy. All patients undertook a personalized rehabilitation program following their surgery.

The mean ASES score improved from 48.1 to 87.8 in the tenotomy group and from 44 to 72.7 in the tenodesis group during the follow-up (p<0.0001).

The tenotomy group had better mean postoperative ASES scores than the tenodesis group (p<0.0001). Positive clinical tests for biceps pathology were noticed more often in patients after LHB tenodesis (p=0.0541). The Popeye deformity occurred more often in the tenotomy group; however, no patient complained on the visual appearance of the arm contour (p=0.0128).

The frequency of night pain decreased in the tenotomy group (p=0.0059). Return to previous sporting activities was more frequent in tenotomy group (p=0.0090).

The authors concluded that an arthroscopic biceps tenotomy was a reproducible, simple procedure, and coupled with a rapid rehabilitation protocol provided very good clinical outcomes, reduced shoulder pain and allowed patients to return to previous sporting activities.

Overall, I enjoyed reading this paper.

This study provides some really useful data, however it is not clear how the patients were selected for each treatment arm? Could the authors explain their surgical decision-making process?

The authors mentioned that the management of chronic biceps tendinopathy required an “individual approach”. What were the clinical criteria that they utilised? This would be helpful to the reader.

Was this a randomised or sequential series? These needs to be clarified.

Was this prospective or retrospective? This also needs to be clarified.

Were any patients excluded from the original cohort of 81?

The tenotomy patients were on average older, and thus the authors should state this more explicitly in their conclusion.

Would the authors also recommend tenotomy in patients younger than 50? What about a 30 year old patient?

Some tidying up of the grammar and English language is also required.

The sentences using the term “enriched” should be changed to “coupled with a” or “augmented by a”

Author Response

Dear Editor,

Thank you for the opportunity to improve and resubmit our manuscript entitled:

“Biceps tenodesis versus tenotomy with fast rehabilitation protocol – a functional perspective in chronic tendinopathy”

The suggestions offered by the reviewers have been immensely helpful. We appreciate all the comments on the manuscript.

We have included the reviewer comments, and responded to them individually, indicating how we addressed each concern and describing the changes we have made. The revised manuscript has been read and approved by all the authors. 

We wish to express again our appreciation for the insightful comments which have helped us significantly to improve our manuscript. 

Yours sincerely,

Jan Zabrzyński

Overall corrections:

  • English language was improved

Reviewer 2:

  • Could the authors explain their surgical decision-making process?

Thank you for the valuable comment. We fully agree with the reviewer and indeed this fact was not particularly described. There were two arthroscopic surgeons, JZ and ŁP, and the first one performed 27 LHBT tenodesis and the second one performed 40 tenotomies of the final follow up population. These two surgeons are trained in mini-invasive arthroscopic procedures, and the methods they chose, are based on their experience. Moreover, during qualifications of patients for an arthroscopic treatment, surgeons explained to the patient, step-by-step, the entire surgical procedure and the possible complications. However, surgeons always told the patients about the preferable arthroscopic technique. Reassuming, the patients were cohort non-randomly into the tenodesis/tenotomy group, and this is the reason why tenodesis patients were statistically younger than those undergoing tenotomy.

We add this issue to the Methods section.

  • The authors mentioned that the management of chronic biceps tendinopathy required an “individual approach”. What were the clinical criteria that they utilised? This would be helpful to the reader.

Thank you for the valuable comment. Biceps tenotomy is actually simple procedure, but it is important, that is not suitable for every patient. There are possible complications, such as: Popeye deformity or 20% decrease in elbow flexion and supination power of the forearm. There is no clear indication which surgical procedure gains a better outcome. From our point of view, the individual approach means the discussion with patient a step-by-step surgical procedure and subsequent rehabilitation protocol. During the qualification of patients for an arthroscopic treatment the surgeon explained the possible complications, however, the preferable arthroscopic technique was always indicated.

We add this issue to the Methods section.

  • Was this a randomised or sequential series? These needs to be clarified.

Thank you for the valuable comment. It was a sequential series. We add this issue to the Methods section.

  • Was this prospective or retrospective? This also needs to be clarified.

Thank you for the valuable comment. It was a retrospective study. We add this issue to the Methods section.

  • Were any patients excluded from the original cohort of 81?

Thank you for the valuable comment. The included population of 81 patients was altered only due to unavailable contact with patient or relocation. There was no additional exclusion.

  • The tenotomy patients were on average older, and thus the authors should state this more explicitly in their conclusion.

Thank you for the valuable comment. We add this issue to the Conclusion section.

“Arthroscopic biceps tenotomy is a reproducible, simple procedure, and – enriched with a rapid rehabilitation protocol – provides very good clinical outcomes, reduces shoulder pain and allows the patient to return to previous sport activities, even in population elder than 50 years.”

  • Would the authors also recommend tenotomy in patients younger than 50? What about a 30 year old patient?

Thank you for the valuable comment. We performed the tenotomy procedure in patients with age 30-74, with good clinical outcomes. The LHBT must be cut parallel to its insertion to the labral complex, to form an extensive biceps stump, which results in its entrapment in the area of the biceps pulley in some patients. Moreover, we perform two incisions obliquely in the LHBT to form an anchor to enhance the entrapment in the pulley. It allowed to avoid the Popeye deformity.

  • Some tidying up of the grammar and English language is also required.

English language and style were edited and improved.

  • The sentences using the term “enriched” should be changed to “coupled with a” or “augmented by a”

Thank you for the valuable comment. We fully agree with the reviewer and “enriched” was changed on “augmented by a”.

Reviewer 3 Report

The authors report on the functional outcome following arthroscopic tenodesis versus tenotomy of the long head of the biceps tendon (LHBT). In addition, patients underwent a personalized postoperative rehabilitation protocoll. A total of 81 patients were included and 67 patients (3 moved house from their previous addressess, 11 unavailable for follow-up) were available for final follow up (mean 27 months) with 40 patients undergoing tenotomy and 27 tenodesis. The authors found that the American Shoulder and Elbow Surgeons score (ASES) improved more in the tenotomy group (p<0.0001) and that those patients returned more frequently to previous sporting activties (p=.0.009). The authors conclude, that LHB tenotomy + their personalized rehabilitation protocal provides good clinical outcomes, reduces shoulder pain and allows for a quick return to sport.

Introduction: well written, comprehenisve.

Experimental section: The study is well designed and approved by the local ethics committee.

What I did not like:

  • I do not really understand why only patients >25 years were included as we have seen chronic tendinitis of the LHB in patients younger than that age - especially with the rapidly increasing group of fitness and lifting (bench press!) addicted teenagers. But this should be fine for this study.
  • Minimum follow-up of only 6 months can be a bit to short, should be higher - but can be accepted for this manuscript.

What I liked:

  • nice photos of postoperative rehabilitation programm

Results: comprehensive, nice diagrams.

- why did a patient aged 30 underwent tenotomy?

Discussion: well written, results are discussed and put in context to the existing literature.

Overall, I really liked to read your paper - the question "tenotomy" versus "tenodesis" (especially in patients aged 40-50) is omnipresent for shoulder surgeons. Your paper adds valuable information to this discussion - especially once again supports that the tenotomy does not result in functional impairment.

Author Response

Dear Editor,

Thank you for the opportunity to improve and resubmit our manuscript entitled:

“Biceps tenodesis versus tenotomy with fast rehabilitation protocol – a functional perspective in chronic tendinopathy”

The suggestions offered by the reviewers have been immensely helpful. We appreciate all the comments on the manuscript.

We have included the reviewer comments, and responded to them individually, indicating how we addressed each concern and describing the changes we have made. The revised manuscript has been read and approved by all the authors. 

We wish to express again our appreciation for the insightful comments which have helped us significantly to improve our manuscript. 

Yours sincerely,

Jan Zabrzyński

Overall corrections:

  • English language was improved

Reviewer 3:

  • I do not really understand why only patients >25 years were included as we have seen chronic tendinitis of the LHB in patients younger than that age - especially with the rapidly increasing group of fitness and lifting (bench press!) addicted teenagers. But this should be fine for this study.

Thank you for the valuable comment. We fully agree with the reviewer and we have a good clinical experience with conservative treatment of the younger patients, using methods, such as: PRP, hyaluronic acid or collagen injections.

  • Minimum follow-up of only 6 months can be a bit to short, should be higher - but can be accepted for this manuscript.

Thank you for the valuable comment. The average period of follow-up in the studied cohort was 27 months (range: 7 to 54 months; SD: 13.2).

  • Why did a patient aged 30 underwent tenotomy?

This patient underwent LHBT tenotomy with concomitant anterior labrum repair. The qualification for surgical procedure was performed by ŁP according to his previous experience. What is more, the postoperative ASES after the 54 months follow up was 100 points and there was no Popeye deformity.

Round 2

Reviewer 1 Report

The study design is inappropriate to answer the question which of the two techniques gives superior clinical outcome.

  • A retrospective analysis does not allow “randomised" treatment decisions.
  • The additional pathologies where mentioned but their influence on the overall result remains unclear.
  • The treatment groups received a different rehab protocol concerning time of immobilisation an active motion, so that is not possible to discriminate, whether the result is influenced by the operative technique or the rehab protocol.

Therefor the manuscript should not be considered for publication.

Author Response

Please check the full document as an attachment.

Thank you for the clarification of your critical issues.

“The study design is inappropriate to answer the question which of the two techniques gives superior clinical outcome.”

According to Friedman et al. (2015), both procedures (tenotomy/tenodesis) result in high patient satisfaction and pain relief. They showed that the patients had similar results with regard to perceived cramping, weakness, and deformity. Although, they indicated that pain at the bicipital groove may be greater for those undergoing a subpectoral biceps tenodesis, however, more refined indications for one procedure over the other still remains elusive. What is more, the authors indicated that the choice of biceps tenodesis vs tenotomy for the younger patient with biceps tendon pathology continues to be decided on through a dialogue between patient and surgeon.

Similar, MacDonald et al. (2020) showed in a total of 114 participants with a mean age of 57.7 years (range, 34 years to 86 years) randomized to undergo either biceps tenodesis or tenotomy that ASES and WORC scores improved significantly from pre- to postoperative time points, with a mean difference of 32.3% (P < .001) and 37.3% (P < .001), respectively, with no difference between groups in either outcome from pre- to postoperative 24 months. Also, they concluded that tenotomy and tenodesis, as a treatment for lesions of the long head of biceps tendon, both result in good subjective outcomes but with a higher rate of Popeye deformity in the tenotomy group.

The results showed by Aflatooni et al. (2020) proved that biceps tenotomy and tenodesis are both viable treatments for proximal biceps tendon pathology, yielding high patient satisfaction in the context of concomitant shoulder surgery. They indicated trends toward greater satisfaction and fewer problems in patients with tenodesis. However, younger patients with tenodesis did report perceived downsides. Alternatively, older patients tended to be more satisfied with both procedures overall. What is more, patients in both groups reported with equal frequency (95%) that they would repeat the procedure. 

According to the authors cited above, in our studied group the mean age was about 55 years and the tenotomy/tenodesis procedures differences were additionally diminished. That is why we hypothesized that tenotomy/tenodesis groups will have similar postoperative results. Nevertheless, after changing the rehabilitation protocols, the difference in outcomes may appear. This is the point from which the treatment groups should be conceived as different rehabilitation protocol groups.

“A retrospective analysis does not allow “randomised" treatment decisions.”

We fully agree with the Reviewer. The study is retrospective case series, which from the definition is just a series of cases, and we then look back to try and find associations between the patients treated with two different operative techniques, similar in the outcome. From this point of view, the patients were randomized to undergo either tenodesis or tenotomy. We understand that the difference between a retrospective case series and a retrospective case-control is that the case series lacks a control group, and then is a much weaker design than case-control. However, the two different rehabilitation protocols were compared between procedures with similar postoperative results. We realize from the risk of bias, and this was underlined in the limitation of the study, in the discussion. Moreover, in the methods section, we indicated the retrospective case series study design, which design was also described.

We realize from the risk of bias what was put in the limitation in the discussion section. Moreover, in the methods section, the retrospective case series study design was also described.

“The additional pathologies where mentioned but their influence on the overall result remains unclear. “

RCT

MRCT

SLAP

Calcium deposits

ACJ disorders

Subacromial bursa disorders

Anterior labrum injury

Omarthrosis

Isolated LHBT pathology

Tenotomy

14

10

5

1

1

5

3

4

2

Tenodesis

8

8

1

0

6

11

0

0

1

As we accented in the discussion section, this kind of tendinopathy is usually associated with concurrent shoulder pathologies, which significantly complicates the diagnostic process and treatment. We also added that this was the main limitation of the study. Some of patients had simultaneously few pathologies and this also may have an impact on the clinical outcomes, however, there were only 3 patients in the studied population with isolated biceps tendinopathy. We are aware, that the tenotomy group had 24/40 (60%) RCTs with massive injuries and the tenodesis group had 16/27 (59%) the same type injuries, but the percentage are almost similar. Moreover, probably the RCTs could have the largest impact on the postoperative ooutcomes.

“The treatment groups received a different rehab protocol concerning time of immobilisation an active motion, so that is not possible to discriminate, whether the result is influenced by the operative technique or the rehab protocol.”

Reassuming, according to the literature and numerous clinical trials, these two operative modalities have similar clinical outcomes, we aimed to alter the rehabilitation protocol after tenotomy procedure, to achieve better functional outcomes, especially >50 years population.

Overall, the additional text and citations were introduced to the: introduction, methods, results, discussion and conclusions sections.
